# Encouraging Sustainable Use of RAP Materials for Pavement Construction in Oman: A Review

Husam Al Dughaishi [1], Jawad Al Lawati [1], Munder Bilema [2], Ali Mohammed Babalghaith [3], Nuha S. Mashaan [4], Nur Izzi Md. Yusoff [5] and Abdalrhman Milad [1,*]

1 Department of Civil and Environmental Engineering, College of Engineering, University of Nizwa, P.O. Box 33, Nizwa P C 616, Ad-Dakhiliyah, Oman; husam@unizwa.edu.om (H.A.D.); jawad.allawati@unizwa.edu.om (J.A.L.)
2 Department of Civil Engineering, University of Benghazi, Benghazi 1308, Libya; munderbilema@gmail.com
3 Center for Transportation Research, Department of Civil Engineering, Faculty of Engineering, University of Malaya, Kuala Lumpur 50603, Selangor, Malaysia; bablgeath@hotmail.com
4 School of Civil and Mechanical Engineering, Faculty of Science and Engineering, Curtin University, Perth, WA 6102, Australia; nuhas.mashaan1@curtin.edu.au
5 Department of Civil Engineering, Universiti Kebangsaan Malaysia, Bangi 43600, Selangor, Malaysia; izzi@ukm.edu.my
* Correspondence: a.milad@unizwa.edu.om

**Abstract:** The Sultanate of Oman has experienced rapid development over the last thirty years and has constructed environmentally friendly and sustainable infrastructure while it continues to find economical alternative resources to achieve the goals of the Oman 2040 vision. The primary concerns are preserving natural resources and reducing the impact of carbon dioxide ($CO_2$ emissions on the environment. This review aims to encourage the sustainable use of reclaimed asphalt pavement (RAP) materials in pavement construction and focuses primarily on employing RAP materials in new pavement projects. Currently, new construction projects utilise a significant percentage of demolished asphalt pavement to save costs and natural resources. The key issue that arises when mixing RAP into new asphalt mixtures is the effects on the mixtures' resistance to permanent disfigurements, such as fatigue cracks, that influence asphalt mixture performance. Numerous studies have assessed the impact of using RAP in asphalt mixtures and found that RAP increases the stiffness of asphalt mixtures, and thus improves rutting resistance at high temperatures. Nevertheless, the findings for thermal and fatigue cracking were found to be contradictory. This review will address the primary concerns regarding the use of RAP in asphalt pavements, and aims to encourage highway agencies and academic researchers in the Gulf countries to develop frameworks for the practical usage of RAP in the construction of sustainable pavement systems.

**Keywords:** asphalt recycling; reclaimed asphalt pavement (RAP); sustainable development; rejuvenators; flexible pavement

## 1. Introduction

Researchers are dedicated to promoting the construction of sustainable asphalt pavements that are energy efficient and require fewer natural resources [1]. Despite the advances in RAP mixture technology and the performance and durability of waste asphalt pavement, waste recycling still poses significant challenges in Oman due to the shortage of disposal sites and the high rate of waste generation [2]. In 2021, Oman had a population of almost four million and generated about 1.6 million tonnes of solid waste, which is more than 1.5 kg per day; the amount of waste generated in Oman is among the highest globally [3]. The waste materials generated in Oman contain a high percentage of recyclables, namely 5% glass, 11% metals, 12% plastics and 26% paper [4]. Most waste materials were disposed of in authorised and unauthorised dumpsites and created health and environmental

problems [5–7]. Recycling waste pavement materials is a sustainable alternative to road maintenance and rehabilitation [8]. Reusing RAP is an economically attractive option in Oman since some areas face a shortage of virgin aggregate and asphalt binder. Road network rehabilitation offers a valuable resource in highway construction [9–11]. The primary reason for using RAP is to remove the pavement materials containing asphalt and aggregates from the existing pavement. The pavement materials are recycled through a milling process that removes the upper pavement layers and replaces them with new pavement, or full-depth removal of existing pavement and reprocessing of the removed pavement [12,13]. Furthermore, the maintenance of road infrastructure produces a considerable amount of RAP that can be recycled without degrading its functionality, as shown in Figure 1. Consequently, it is essential to evaluate the multi-recycling capacity of the asphalt mixtures.

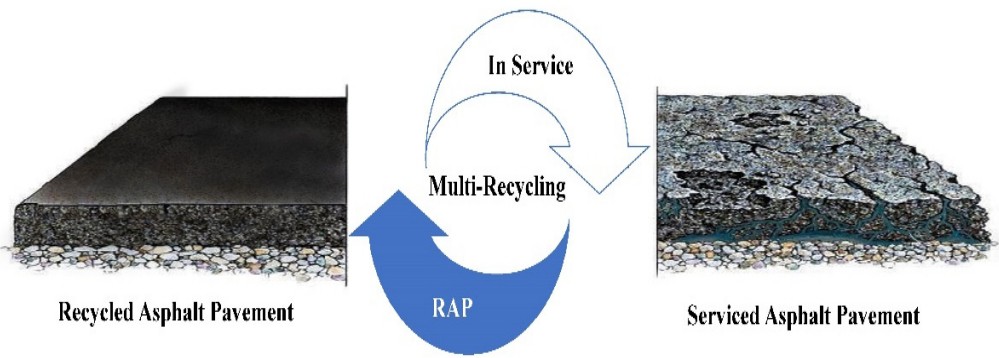

**Figure 1.** RAP recycling process.

RAP is suitable for re-construction or re-surfacing roads. RAP contains well-graded fine aggregates covered with asphalt cement and must be properly crushed and categorised. The four most frequently used processes for recycling RAP conserve natural resources and make pavement construction sustainable [14,15]. RAP is a suitable substitute for virgin materials because it reduces the use of virgin aggregates, thus reducing the quantity of new overpriced asphalt binder required for asphalt paving mixtures [16]. Three critical factors ensure the successful use of RAP, (i) cost-effectiveness, (ii) good performance, and (iii) environmentally friendly. To meet these requirements, the Federal Highway Administration of the US Department of Transportation (FHWA) has set the following objectives to encourage asphalt pavement recycling [17]:

- Gain utilization of recycled materials on highways in the most economical and practical range, likely with equal or improved public accomplishment.
- Encouragement of the purpose of RAP is intended to apply the considerable economic, environmental, and engineering influences to asphalt pavement recycling.

This review aims to encourage experimentation with recycled RAP mixtures; determine the performance of the asphalt mixtures, particularly concerning the cost, energy consumption, and materials; and evaluate the RAP mixture design methods adopted in the Gulf countries with the recommendation of guidelines to incorporate the RAP mixture in the newly approved designs.

## 2. The Use of RAP in the Gulf Countries

Countries across the globe are exploring the use of sustainable and environmentally friendly materials, and RAP is currently a trendy new pavement material in road construction. The Gulf countries are transitioning to sustainable practices. For instance, the United Arab Emirates (UAE) uses RAP mixes in its current projects (such as the Kadra-Shawka road) without clear guidelines on cost reduction, while ensuring good pavement performance [18,19]. In 2020, Hasan et al. [20] investigated the utilisation of RAP in a segment of a 3.5 km highway. However, the study compared using RAP with Warm Mix Asphalt

(WM) and Hot Mix Asphalt (HMA) with recycled top-distressed asphalt pavement as natural aggregates and the environmental benefits of these alternatives for a life cycle of 30 years [21]. The study revealed that using RAP in the WMA by using about 15% of RAP content in the binder and wearing courses coupled with WMA resulted in significantly lower environmental impacts across all indicators [22].

Furthermore, at some stage in the rehabilitation, using a milled wearing course demonstrated that WMA has the capability of using high RAP content in the production mix. About 85% of in-plant WMA recycling of milled RAP was assessed and compared to virgin HMA asphalt. In Oman and the Gulf countries, RAP recycling is one of the best road maintenance options [9]. Road re-construction produces large amounts of RAP, which can be recycled into new asphalt pavement mixes. However, RAP is infrequently used in Oman because of the lack of expertise. Laboratory evaluation of cement stabilised RAP and RAP-virgin aggregate blends as base materials showed that the optimum moisture content, maximum dry density, and strength of RAP generally increase with the addition of virgin aggregate and cement, while longer curing time increases the yield strength. It is not feasible to use 100% RAP aggregate as a base material unless it is stabilised with cement [10].

In Egypt, the increasing demand for using HMA mixtures is produced merely from virgin material converted to green asphalt pavement worldwide [23]. However, recently, the existing waste RAP material, about 4 million tonnes annually, affecting landfills has led to environmental impacts by reducing energy consumption [24]. In 2015, El-Maaty & Elmohr [25] determined the mechanical properties and durability of dense-graded HMA mixtures by incorporating RAP materials. The results showed that a replacement ratio of 50% to 100% produced better durability, mechanical properties, and stripping resistance. Alwetaishi et al. [18] investigated the effects of incorporating 0%, 30%, 60% and 90% RAP and found that 90% RAP is the optimum percentage for road construction. The study similarly found that asphalt concrete mix with 90% RAP has the ideal energy efficiency; however, it is only suitable for building applications in regions with cold climates.

Compared to other recycled materials used globally, RAP is the most frequently recycled material [18]. In 2013, Sultan developed a technique to enhance the mechanical properties of RAP by mixing 40% RAP with virgin material to produce RAP with the minimum required mechanical properties for road construction. Analysis of the new mix showed that using RAP could save around 39% of the total pavement cost, hence providing Iraq with a cost-effective, environmentally friendly, and sustainable road construction technique [19]. RAP is used as a substitute for virgin aggregates throughout the globe. The UK, the USA, Japan, Canada, and other developed countries have well-established procedures, guidelines, and standards to classify and reuse recycled asphalt materials. The Gulf countries, the Middle East, and other developing countries do not have adequate standards, guidelines, and efficient procedures for determining the reclaimed asphalt materials [26–29].

## 3. Classification of RAP Recycling Methods

The in situ recycling and central plant recycling methods for RAP are: cold central plant recycling (CCPR), hot in-place recycling (HIR), cold recycling, or cold in-place recycling (CIR), and full depth reclamation (FDR). In the in-situ recycling method, RAP modification is completed at the construction site, whereas in the central plant recycling method, the RAP modification is accomplished at a plant away from the construction site. Figure 2 shows the classification of RAP based on the recycling technique.

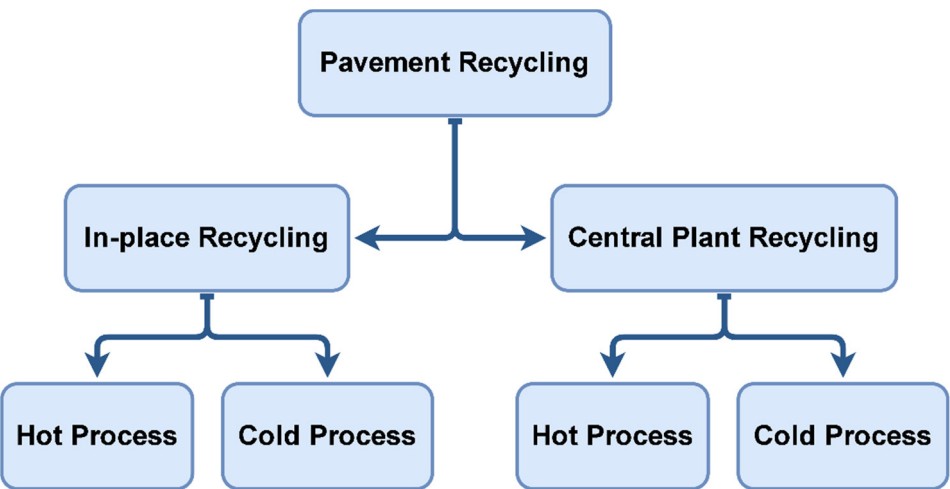

**Figure 2.** Classification of asphalt pavement recycling methods.

RAP recycling is classified as hot or cold recycling, depending on the use of heat. Cold recycling uses cutback or emulsion as recycling agents [30,31]. The recycling methods are also determined by the removal depth of the old pavement. Surface recycling involves removing and relaying the upper layer of the pavement, while full-depth reclamation requires removing and relaying all pavement layers, including the base [32]. Another method, hot-in-place recycling, requires heating and scarifying the pavement to the required depth, and depending on the characteristics of the milled RAP materials, this is followed by adding bitumen and new aggregates. The resulting mixture is then laid and compacted. This method reduces transportation costs, causes minor traffic disruption, and is less time consuming, but requires bulky machinery [33].

On the other hand, the CIPR approach does not require on-site heat application and uses cutback or emulsion as a binder. This method requires adequate time to cure the freshly laid layer. Additives, such as cement, quick lime or fly ash, can be used to reduce the emission of harmful gases [34,35]. Hot central plant recycling requires adding a bituminous binder and new aggregates to the RAP material in a hot mix plant away from the site. The properties and performance of the mixtures produced using this method are similar to the virgin hot mixture [35] because of the better-quality control in the central plant recycling method [31].

It is essential to store RAP material properly at construction sites that do not have sufficient storage because the RAP material is vulnerable to moisture [36,37]. Cold central plant recycling does not involve heat application at the plant and instead uses cutback or emulsion as a binder. The mixing time is critical because overmixing may prematurely break the emulsified binder. However, undermixing may cause an insufficient coating of the aggregates [30]. Table 1 summarises the research sought to determine the best RAP methods for pavement construction and rehabilitation.

**Table 1.** Previous research on RAP production methods.

| Researchers [Refs] | Country | Year | Cold Recycling | | | Hot Recycling | |
| | | | CIR | CCPR | FDR | HIR | HMAR |
|---|---|---|---|---|---|---|---|
| Cross et al. [38] | The U.S.A | 2010 | ✓ | | | | |
| Kamran et al. [39] | Pakistan | 2012 | ✓ | | | | |
| Apeagyei et al. [40] | The U.S.A. | 2013 | ✓ | | | | |
| Stimilli et al. [41] | Italy | 2013 | | ✓ | | | |
| Feisthauer et al. [42] | Canada | 2013 | | | ✓ | | |
| Hafeez et al. [43] | The U.S.A. | 2014 | | | | ✓ | |
| Bhavsar et al. [44] | India | 2016 | ✓ | | | | |
| Turk et al. [45] | Slovenia | 2016 | ✓ | | | | |
| Noferini et al. [46] | Australia | 2017 | ✓ | | | | |
| Zhao & Liu [47] | The U.S.A. | 2018 | | | | | ✓ |
| Graziani et al. [48] | Italy | 2018 | | | ✓ | | |
| Vázquez et al. [49] | Spain | 2018 | ✓ | | | | |
| Bowers et al. [50] | The U.S.A. | 2019 | | ✓ | | | |
| Gonzalo et al. [51] | Spain | 2020 | | | ✓ | | |
| Jovanović et al. [52] | Serbia | 2021 | | | | | ✓ |
| Iwański et al. [53] | Poland | 2022 | | | ✓ | | |

## 4. RAP Standards and Mixed Design

Recently, the Oman government encouraged agencies to use RAP. The government regulations or standards for the mixing ratio, RAP analysis, and assembly/testing procedures should be established for the public and private contractors for the use of RAP [10,54]. Taha et al. [9] performed numerous tests on the feasibility of using 100/0, 80/20, 60/40, 20/80, and 0/100% RAP with virgin aggregates in road base and sub-base following the AASHTO T180 and AASHTO T193 [55]. The results showed that the maximum dry density decreased with higher RAP aggregate percentages and the California bearing ratio (CBR) increased with higher virgin aggregate contents in the blend, but the CBR decreased when using RAP as a complete replacement. Furthermore, the guideline established by the Superpave Mixtures Expert Task Group states that the design of HMA with RAP is based on a three-tier system [56,57]. Up to 15% of RAP can be utilised without changing the virgin binder grade of those chosen for the project conditions and locations. A 15 to 25% RAP content reduces the virgin binder's low- and high-temperature by one grade, causing the ageing binder's stiffening effect.

Lee et al. (1999) [58] examined the mechanical and rheological properties of blended asphalts containing RAP binders. However, the binders used were PG 58-28 and PG 64-22. When the amount of RAP in HMA exceeds 25%, blending charts are used to determine the suitable percentage of RAP with a given virgin binder. When using a blending chart, the RAP binder must be extracted, recovered, and tested following the specifications and guidelines presented in Table 2. Several factors determine the quantity of RAP in the new mixture, such as the specification limits for the type of mixtures; plant type; gradation; aggregate consensus properties; binder properties; the drying, heating and exhaust capacity of the plant; the moisture content of the RAP and the virgin aggregates; temperature of the virgin aggregate must be the superheated; and the ambient temperature of the RAP and virgin aggregate [59,60]. The limiting factors could be related to the material or the production process. An example of a production-related factor is the plant's capacity to dry and heat the virgin aggregates and the RAP, where more energy is used to heat and dry the

materials when the ambient temperature is low or when the material has a high moisture content. These factors influence the production rate of HMA.

**Table 2.** The RAP recycling formulas for development and estimation [56–64].

| Equation No. | Calculation Model | Purpose of Use |
|---|---|---|
| 1 | $$Mc = \frac{RAP^w - RAP^d}{RAP^d} \times 100$$ where: <br> $Mc$ = Moisture content expressed as a percentage <br> $RAP^w$ = Weight of aggregate in stockpile condition <br> $RAP^d$ = Weight of aggregate in SSD condition | Calculate the Moisture content of RAP |
| 2 | $$Gse = \frac{100 - P_b}{\frac{100}{Gmm} - \frac{P_b}{G_b}}$$ where: <br> $Gmm$ = Maximum specific gravity of the mixture <br> $G_b$ = Specific gravity of the asphalt cement <br> $P_b$ = Asphalt cement content as a percentage of the total mixture | Calculate the effective specific gravity of RAP |
| 3 | $$Gsb = \frac{Gse}{\left(\frac{Pba \times Gse}{100 \times Gb} + 1\right)}$$ where: <br> $Gse$ = Effective specific aggregate gravity of aggregate <br> $Gsb$ = Bulk Specific Gravity of the aggregate <br> $Pba$ = Asphalt absorption of the aggregate <br> $Gb$ = Specific gravity of the asphalt cement | Calculate the bulk specific gravity of RAP |
| 4 | $$\%RAP = \frac{T_{blend} - T_{virgin}}{T_{RAP} - T_{virgin}}$$ where: <br> $T_{virgin}$ = Critical temperature of the virgin asphalt binder <br> $T_{blend}$ = Critical temperature of the blended asphalt binder <br> $\%RAP$ = Percentage of RAP to be used; and <br> $T_{RAP}$ = Critical temperature of the recovered RAP binder. | Calculate the percentage of RAP |
| 5 | $$M_{dryRAP} = \frac{M_{RAPAgg}}{(100 - P_b)} \times 100$$ where: <br> $M_{dryRAP}$ = Mass of dry RAP <br> $M_{dryAgg}$ = Mass of RAP aggregate and binder <br> $P_b$ = RAP binder content | Calculate the mass of dry RAP |
| 6 | $$T_c(High) = \left(\frac{\log 1 - \log G_1}{a}\right) + T_1$$ where: <br> $T_c$ (high) = at which $G^*/\sin\delta$ is equal to 1.00 kPa <br> $G_1 = G^*/\sin\delta$ at temperature $T_1$ <br> $T_1$ = Critical temperature <br> $a$ = slope of the stiffness-temperature curve | Determine the critical high temperature |
| 7 | $$T_c(High) = \left(\frac{\log 2.2 - \log G_1}{a}\right) + T_1$$ where: <br> $T_c$ (high) = At which $G^*/\sin\delta$ is equal to 2.2 kPa <br> $G_1 = G^*/\sin\delta$ at temperature $T_1$ <br> $T_1$ = Critical temperature <br> $a$ = Slope of the stiffness-temperature curve | Determine the critical high temperature based on RTFO |
| 8 | $$T_c(Int) = \left(\frac{\log 5000 - \log G_1}{a}\right) + T_1$$ where: <br> $T_c$ (Int) = At which $G^*\sin\delta$ is equal to 5000 kPa <br> $G_1 = G^*/\sin\delta$ at temperature $T_1$ <br> $T_1$ = Critical temperature <br> $a$ = Slope of the stiffness-temperature curve | Determine the critical intermediate temperature |

**Table 2.** *Cont.*

| Equation No. | Calculation Model | Purpose of Use |
|:---:|:---:|:---:|
| 9 | $T_c(S) = \left(\frac{\log 300 - \log S_1}{a}\right) + T_1$<br>where:<br>$T_c$ (S) = Critical low temperature<br>$S_1$ = S-value at temperature $T_1$<br>$T_1$ = Critical temperature | Determine the critical low temperature |
| 10 | $T_c(m) = \left(\frac{0.3 - m_1}{a}\right) + T_1$<br>where:<br>$T_c$ (m) = Critical low temperature<br>$m_1$ = m-value at temperature $T_1$<br>$T_1$ = Critical temperature<br>a = Slope of the stiffness-temperature curve | Determine the critical low temperature |
| 11 | $T_{blend} = T_{virgin}\,(1 - \%RAP) + T_{RAP} \times \%RAP$<br>where:<br>$T_{virgin}$ = Critical temperature of the virgin asphalt binder<br>$T_{RAP}$ = Critical temperature of the blended asphalt binder<br>%RAP = Percentage of RAP to be used<br>$T_{RAP}$ = Critical temperature of the recovered RAP binder. | Determine the critical temperature of the blended asphalt binder |
| 12 | $T_{virgin} = \frac{T_{blend} - (\%RAP \times T_{RAP})}{(1 - \%RAP)}$<br>where:<br>$T_{virgin}$ = critical temperature of the virgin asphalt binder<br>$T_{blend}$ = critical temperature of the blended asphalt binder<br>%RAP = percentage of RAP to be used; and<br>$T_{RAP}$ = critical temperature of the recovered RAP binder. | Determine the critical temperature of the virgin asphalt binder |
| 13 | $X_{RAP\ binder\ (s)} = \frac{RAP\ content\ (\%)}{100} \times \frac{RAP\ binder\ content\ (\%)}{Asphalt\ mix\ binder\ content\ (\%)}$<br>where:<br>$X_{RAP\ binder(s)}$ = RAP binder volume fraction<br>$X_{rejuvenator}$ = Rejuvenator oil volume fraction | Calculate the binder content for the asphalt mix |
| 14 | $D = \frac{\log\frac{PEN}{A}}{B}$<br>where:<br>A = RAP percent binder content<br>B = RAP percent in mixture<br>D = Rejuvenator dosePEN = the penetration $\times$ 0.1 mm | Calculate the rejuvenator dosage |
| 15 | $E = \frac{0.5Pd}{t}$<br>where:<br>E = energy (lb.in/in)<br>P = ultimate load at failure<br>d = specimen vertical deformation at the ultimate load (in)<br>t = specimen thickness (in) | Calculate the absorbed energy |
| 16 | $PER = \frac{E_{conditioned}}{E_{control}}$<br>where:<br>PER = Percent of absorbed energy<br>$E_{conditioned}$ = Average level of absorbed energy for conditioned specimens<br>$E_{cotrol}$ = Average level of absorbed energy for control specimens | Calculate the percentage of absorbed energy |

## 5. RAP Performance Analysis

Even though RAP was first used in 1915, it was seldom used until the early 1970s, when the price of asphalt binder increased due to the Arab oil embargo. In 1973, RAP was used in civil engineering despite the lack of knowledge about the optimal RAP ratio and the performance of asphalt mixes containing RAP [11,65]. Currently, virgin asphalt mixes contain high percentages of RAP to reduce cost and ensure a more sustainable construction by recycling old asphalt pavements [66]. Researchers across the globe sought to understand

the technical aspects of RAP, its performance with thermal and fatigue cracks, and its resistance to continuous deformation (rutting strength) [67] since rutting is a critical factor in RAP performance [68,69]. Pradyumna et al. [70] found that, relative to virgin mixes, the asphalt mixes containing 20% RAP had better rutting resistance. The addition of RAP increased the rutting resistance as the RAP stiffened; the penetration and softening points for the RAP binder were 39 (0.1 mm) and 62 (°C). The results of previous tests showed less ageing in RAP binders. Other studies showed that mixes with 50% and 100% RAP enhanced the asphalt pavement performance and that higher RAP mixing ratios could improve the rutting strength and resilient modulus [71]. Boriack et al. [68] conducted four trials using 0%, 20%, 40%, and 100% RAP and found that higher RAP percentages increased the stiffness and reduced the permanent deformation.

Analysis of the mixes containing 0%, 5%, 10%, and 15% RAP showed that mixtures with higher RAP percentages have higher rutting resistance due to the lower creep strain. The continuous loading on the asphalt layer imposed higher stress and strain on the bottom layer and caused fatigue cracking [72]. Tabakovic et al. [73] found that incorporating 30% RAP improved the fatigue resistance and the mechanical properties. Pradyumna et al. showed that adding 20% RAP increased fatigue by 67.2% [70]. Sunil et al. [74] tested three different stress ratios of 60%, 70%, and 80% to determine the fatigue resistance. When adding different RAP ratios, they found that higher RAP percentages reduced fatigue resistance as the number of cycles to failure decreased with the higher RAP percentages.

Chaitanyaa et al. [75] investigated the results of incorporating 0%, 15%, 25%, 35%, and 50% RAP and found that the mixtures containing up to 35% RAP improved fatigue resistance by increasing the pavement stiffness and strain while reducing fatigue. The continuous loading on the asphalt pavement could cause thermal cracking in cold temperatures; however, there is no risk of fatigue cracking occurring in Oman because the daily average temperature is about 20 °C during winter [76]. Johnson et al. [77] performed an indirect tensile test (IDT) and found that the creep stiffness increased at lower temperatures, and the crack resistance and fracture resistance reduced with higher RAP ratios. Another study showed that higher RAP percentages reduced the fracture energy and thus did not meet the minimum limit or recommended levels in the USA. The average RAP percentage in a blended asphalt mixture is between 12 and 15% [78]. Researchers in the USA and Canada have shown that the performance of blends with at least 30% RAP was similar to that of blends containing virgin aggregates [79–82]. The ideal percentage of 30% RAP replacement improved all asphalt mixture properties.

## 6. Characteristics of Incorporating RAP

The primary concern when analysing RAP properties is the performance of recycled asphalt (RA) mixtures [83]. Asphalt binders lose some of their original properties during the ageing process. Figure 3 shows that a complete blending occurred after the new binder was added to the RAP and the further distribution during storage resulted in almost complete mixing.

This highlights the importance of exploring the appropriate mixing controlling processes between recycled and virgin aggregates. Studies have shown that small percentages of up to 20% aged asphalt binders did not enhance or affect the properties of RAP or virgin aggregates [84]. However, higher aged asphalt binder content affected the mixture's performance and improved the binder grade. The rectification and balancing of the asphalt binders in terms of missing elements can lead to early ravelling and cracking since the binder serves as the glue. Thus, lower asphalt content can also mean that the mix is harder to compact. Recycling agents comprise softening agents and rejuvenating agents. The softening agents, such as asphalt flux oil, slurry oil, and lube stock, reduce viscosity and thus the stiffness of RAP binders. Rejuvenating agents contain lubricant and extender oils that have high percentages of constituents that restore the physical and chemical properties of RAP [85].

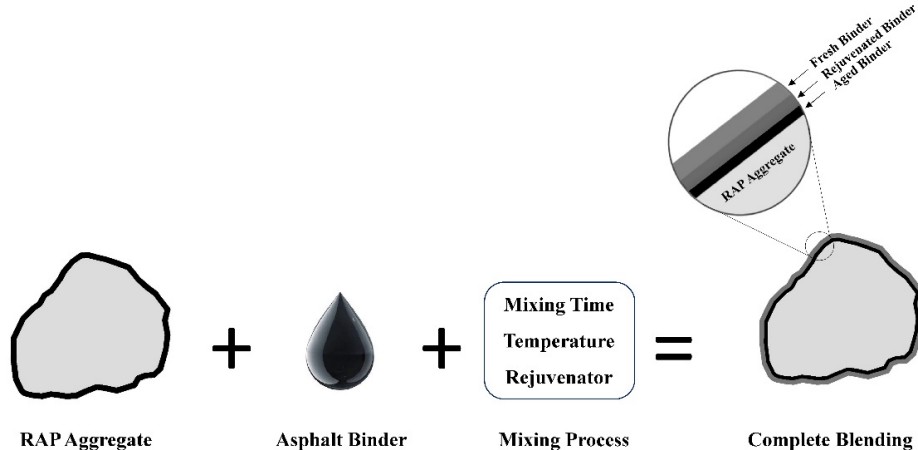

**Figure 3.** A complete blending of the virgin binder and RAP binder.

## 7. Case Studies for RAP Application

Road maintenance and rehabilitation are costly, and researchers have been looking for methods to reduce the cost of maintenance and rehabilitation works. One way to minimise costs associated with the maintenance and rehabilitation works is recycling pavement materials using various recycling methods, and each country has its preferred recycling methods.

### 7.1. Case Studies in the United States of America

The Virginia Department of Transportation (VDOT) finished a notable asphalt recycling scheme that marked the first run through FDR in 2011. Both CCPR and CIR have been utilised in the US interstate framework. The CCPR and CIR blends were produced using hydraulic cement and foamed asphalt as the stabilising agents. After finishing the asphalt recovery scheme, the building characteristics of the CCPR and CIR blends were resolved in the lab from field cored examples. VDOT led this investigation because the organisation was considering utilising a solitary arrangement of development details for both CCPR and CIR materials if the building characteristics of the two procedures were observed to be comparable [40]. In the study by VDOT, it was determined that the dynamic modulus of field-cured and field-produced recycled asphalt materials was equal to 24 projects in Canada and the United States. The study determined the effectiveness of using foamed asphalt-stabilised RAP from full-depth reclamation (FAS-FDR) as a base material for flexible pavements. The measurable test and perception of the developed ace bends showed that the three asphalt recycling methods: full-depth reclamation, cold in-place recycling, and cold central-plant recycling, had comparable dynamic modulus values [86]. Romanoschi et al. (2004) determined that the adequacy of the utilisation of frothed asphalt balanced the recovered asphalt from full-profundity recovery (FAS-FDR) as the base material for adaptable asphalts. The trial by the Civil Engineering Infrastructure Systems Laboratory at Kansas State University comprised the development of four asphalts, one with a 9 in. Regular Kansas AB-3 granular base and one each with 6, 9, and 12 in [87].

Saskatchewan is experiencing a considerable increase in business truck traffic due to grain transportation legitimization, the union of the provincial grain lift framework, country monetary enhancement, and the extension of asset ventures. Therefore, the Saskatchewan Department of Highways and Transportation is investigating the feasibility of using CIR and FDR cementitious adjusted to reinforce the thin asphalts [88].

The Nevada Department of Transportation (NDOT) has been using CIR and FDR for over 20 years, which has made it possible for NDOT to save more than $600 M in contrasted and finished recreation costs in recent years. NDOT uses a proactive pavement management system (PMS) to organise its asphalt protection schemes [89]. An overview was conducted to get relevant data concerning current FDR practices in Minnesota. This

review utilises mechanics-based material testing systems, MnPAVE recreations, an LCCA, and a low-temperature execution investigation to assess FDR materials. These outcomes will aid in the management of the improvement of execution-based particulars, for example, the method utilised for other asphalt materials such as blends and asphalt fasteners [89]. The University of Illinois at Urbana-Champaign explored an in situ recycled material, a trademark that incorporates the capability of asphalt blends for perpetual disfigurement, weariness breaking, and low-temperature splitting, and the impact of a unique strategy for hot set-up recycling and revival on asphalt fastener rheological properties. Findings revealed that the solidness of the asphalt blends in the wake of recycling had expanded in contrast to that before recycling [90]. The utilisation of FDR and HIR by organisations was conveyed across the United States. Information on the degree, sort, and undertaking highlights of FDR, CIR and HIR was gathered through an online review of state offices and temporarily set up recycling workers [91]. Organisational and contractual worker reactions were considered, and rules have been laid for the most suitable traffic, atmosphere, and geometric highlights of tasks. Findings revealed that FDR tasks could be developed in all areas, whereas HIR and CIR require caution when built into wet, hot, and chilly atmospheres [92].

Unlike other states, Alaska routinely incorporates RAP into WMA for pavement construction. Generally, RAP is used in the construction of roadways and airport runways. Some companies in Alaska specialise in processing RAP for future projects. Alaska is experiencing depletion of high-quality natural resources, and many agencies are increasing the utilisation of RAP in their projects [47].

### 7.2. Case Studies in European Countries

For the last 30 years, European countries have used recycled RAP asphalt pavement in new asphalt production. On the other hand, at least one-third of RAP ends up as non-asphalt concrete by optimising the application process [22]. Many rehabilitation and maintenance works were done, but those were somewhat expensive, such as in Denmark, France, Germany, Italy, Netherlands, and Spain [93]. In 2022, Iwaski et al. conducted CIR methods that rely on a three-component hydraulic binder that was blended with 40% cement, 20% hydrated lime, and 40% cement by-pass dust. However, the RAP quality can be adapted to the needs of road-building practice [53].

Stimilli et al. investigated the potential of using a CIR asphalt mixture as a base course for an Italian roadway. The mixture was produced in a central plant using high-reclaimed asphalt (RA) and utilised to develop two experiments along an Italian roadway. Specifically, a unique mixing system, including bituminous emulsion and water vapour has been tested [41].

To achieve the key objectives of the 2013–2020 Spanish strategy for science, technology, and innovation, which emphasizes environment and climate-friendly, resource-efficient in providing a safe and comfortable European transport system, the pavements built must be built using a CIR technique [94]. Moreover, the increasing awareness of environmental impacts has made CIR the preferred method for constructing and rehabilitating pavement.

### 7.3. Case Studies in Asian Countries

Continuous traffic load caused distress, including segregation of aggregates, severe crocodile cracks, and rutting, on the Vadodara Halol Road in India. For instance, in Asian countries such as Japan, there are 1200 asphalt plants spread across the country. It is found that the recycling ratio of the asphalt mixture and cement concrete is higher [94]. Moreover, several researchers provide their methodology and procedure to attempt the use of pavement layers in Asian countries, such as China, Bangladesh, India, and Nepal, which are also racing to use RAP as a pavement construction material, particularly as a base and subbase material [95–97]. Therefore, the CIR method is then suggested to be used to reduce rehabilitation and maintenance costs. In 2016, Bhavsar et al. [44] proved the cost-effectiveness of the CIR method. By reviewing all three case studies presented above,

the general trend for encouraging and practising recycled RAP mixtures and having a clear picture of the performance characteristics of the asphalt mixtures, especially in terms of cost, energy, and material in the Gulf countries, as well as methods of RAP based on guidelines recommending the incorporation of RAP mix designs in the road government sector becomes apparent.

## 8. Conclusions

Roadways are the foundation of any country's social development and economy, which is relied upon to be constructed to last with good road network management. The Oman 2040 vision for future smart cities could achieve its objective of constructing eco-friendly roads and maintaining and upgrading low-volume roadways by using recycled materials such as RAP and other end-of-life materials. Furthermore, using recycled materials provides a long-term solution for reducing the disposal of end-of-life materials in the landfill and reduces construction costs and the dependence on virgin materials. The findings of this review are as follows:

- The design and manufacture of a high-quality RAP mixture is more challenging than conventional asphalt and thus should be done by experienced pavement engineers.
- The RAP mixture design is dependent on the standards, scope of application and the impact of RAP technologies on the environment and economy.
- For the Oman scenario, this review guideline is for engorgement RAP applications for flexible pavement layers using WMA and HMA.
- Assembling homogeneousness in RAP material quality check is essential and should be carried out from the commencement phase of old pavement reclamation to its final paving.
- The maintenance and rehabilitation costs and the environmental influence, as RAP methods use recycled materials, finally reducing waste established along the previous research works stated.
- The CIR method is often used for rehabilitation and construction work because it consumes less energy, emits fewer greenhouse gases and is more cost-effective than other on-site pavement recycling methods.
- The RAP application encouragements as a value are used as a granular base material for roadway, airport runways, shoulders, engineered fill, and culvert backfill.
- This study suggests that using RAP can be used by adding a percentage less than 20% because it is less complex and does not need additions of asphalt binder. Thus, it will be an encouraging step toward improving environmental sustainability constructions in Oman.

**Author Contributions:** A.M.: Conceptualisation, Writing—Original draft, Methodology, Investigation, Formal analysis; J.A.L.: Supervision, Reviewing, Resources; H.A.D.: Supervision, Review, Fund acquisition; M.B.: Writing, curation; A.M.B.: Validation, Fund acquisition; N.S.M., A.M.: Fund acquisition, Visualisation; N.I.M.Y.: Data curation, Fund acquisition. All authors have read and agreed to the published version of the manuscript.

**Funding:** The University of Nizwa paid for the Article Processing Charges (APC).

**Institutional Review Board Statement:** Not applicable.

**Informed Consent Statement:** Not applicable.

**Data Availability Statement:** All data used in this research is provided upon request.

**Acknowledgments:** The authors would like to thank the University of Nizwa for its financial support.

**Conflicts of Interest:** The authors have no conflict of interest to declare.

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
