# Peer review of "Encouraging Sustainable Use of RAP Materials for Pavement Construction in Oman: A Review"

_recycling, doi:10.3390/recycling7030035_

Round 1
Reviewer 1 Report
The subject matter is interesting, and the research is planned and performed correctly. However, according to the reviewer, there are some problems existing in this paper which the authors must pay attention to deal with.
- Purpose of this paper is to encourage sustainable utilization of the reclaimed asphalt pavement (RAP) materials in Oman's pavement constructions. More current researches or applications of recycled mixtures in Oman should be given.
- Abstracts can be more concise by removing some secondary information. Such as “The first usage of RAP was in 1973, nonetheless in small quantity, since there was a lack of perception on the significance of their application on the performance of asphalt mixtures.”
- Only some important and significant conclusions could be revealed in the conclusion section.
Author Response
Thank you for your positive, fruitful comments and suggestions, which have improved our manuscript's quality. Please find below is the revision report for your attention and perusal.

Reviewer 2 Report
Before making any recommendations for a interesting scientific article "Towards Encouragement Sustainable Utilization of the RAP Materials for Pavement Construction in Oman: A Review", I would like to present the following statements on the topic. Based on my long-term research and transfer profile in the field of holistic perception of the issues of pavement engineering, I consider the evaluated scientific article to be extremely topical and fully convergent with the following author's research and educational premise. Pavements should be designed, built, managed, maintained, recycled (decomposed) at a reasonable price, in reasonable quality, respecting the relevant requirements of users, residents and sustainable development. One such minimization measures in the field of pavemens may be the including sustainable utilization of the reclaimed asphalt pavement (RAP). Personally, I consider the article under review to be a valuable contribution to Sustainable Pavements of Persian Gulf countries. Kibert (1994) laid down the foundation for Sustainable construction (SC) practice, and established SC around resource minimization and reuse, use of renewable and recyclable resources, and minimizing carbon footprint. Vanegas and Pearce (2000) presented SC based on resource depletion and degradation, impact on built environment and human health, and Pulaski (2004) presented a comprehensive approach towards sustainability in construction operation. Sustainable construction tools and standards were first developed for buildings. The first set of assessment/rating concepts and tools in the US were provided by the: Green Building System (GBS (USGBC LEED); the origin of these tools can be traced back to UK´s Building Research establishment (BRE) environmental assessment method (BREEAM) (BRE 2006; BREEAM 2006). Subsequently other “systems“ were developed: Canada: Green Globe, Japan: CASBEE (Comprehensive Assessment System for Built Environment Efficiency), Singapore: BCA (The Building and Construction Authority) Greenmark, China: Green Olympics Building assessment System (GOBAS), Australia: Building greenhouse rating (ABGR), India: TERI-GRIHA (Green Rating for Integrated Habitat Assessment). Presented, in the conditions of Europe environmentally progressive technology of municipal waste recovery is in full compliance with Although differing in scope, the currently agreed common goals of SC are as follows: reduce carbon footprint, ecology and environmental protection, healthy indoor and outdoor environment, resource efficiency, conserve resources (land and raw materials). In the case of pavement engineering, it is mainly about saving asphalt, aggregates and energy in asphalt and cement concrete roads. Personally, I am pleased that the rich countries of the Persian Gulf are also aware of this fact, as evidenced by the assessed contribution.
Mandatory requirements:
LNSA (Line Number of Scientific Article) 25-26... The first usage of RAP was in 1973, nonetheless in small quantity, since there was a lack of perception on the significance of their application on the performance of asphalt mixtures… It is necessary to specify whether the first usage is meant in the world, Persian Gulf, Oman…
LNSA 43… green asphalt pavements... This term is not used in Europe, so I would recommend using the term sustainable asphalt pavements or putting the term in quotation marks.
LNSA 186-187…AASHTO T180 and 186 AASHTO T193... I recommend listing these regulations in the bibliography, respectively at least give their names
LNSA 179... Table 1. shows the past research on various RAP methods... The table name should start with a capital letter.
LNSA 224-226… The results carried out by Pradyumna et al. [70] showed that mixing asphalt with 20% RAP could anticipate lower rutting depth when it is compared with virgin mixes, and the addition of RAP in the mixing ratio could increase the rutting resistance as RAP becomes stiffer… I dare to disagree with this general statement, it is necessary to specify more detail the compared asphalt mixture (with virgin mixes).
LNSA 286…7.1. A case study in America... You need to change the title of the chapter in the "in America" section.
LNSA 373-401… The conclusions need to be reworked, the abstracted knowledge must also be more details quantified numerically. I dare to disagree with some of the general findings. For example: Several researchers have proven that the operating characteristics of RAP methods can be at least tantamount to new asphalt pavement structures ... is cheaper than other methods for all the recycling processes at the site ... These statements may apply in precisely specified building and climatic conditions, but their general validity is highly debatable. It would also be appropriate to indicate a possible continuation of research in Gulf countries in the field of applications of RAP.
LNSA 420…. Zafar, S. Waste Management Outlook for the Middle East. In The Palgrave Handbook of Sustainability; Brinkmann, R., 419 Garren, S., Eds.; Palgrave Macmillan: Cham, Switzerland, 2018; pp. 159–181... A dot needs to be added at the end of this reference, this also applies to lines: 433, 463, 477, 482, 484, 486, 488, 490, 557, 562, 575, 585, 587, 592, 597, 601.
LNSA 569-570... Abo-Qudais S, Ibrahim A, Al-Ramahi EJJoG and Engineering T 2016 Utilizing Reclaimed Asphalt Pavement in Asphalt Mixtures: Laboratory Performance and Environmental and Cost Impacts 2 ... I recommend quoting the format in the following form. Abo-Qudais, S., Ibrahim, A., & Al-Ramahi, E. Utilizing (2016). Reclaimed Asphalt Pavement in Asphalt Mixtures: Laboratory Performance and Environmental and Cost Impacts. Journal of Geotechnical and Transportation Engineering, 2 (1).
Facultative recommendations:
LNSA 37…in Gulf countries... The name Persian Gulf is not used in the European area, it might be appropriate for a European reader to mention Gulf countries (hereinafter also Gulf) for the first time.
LNSA 42-86... Until the 1st Introduction, I would like to recommend considering the inclusion of references to the following articles. At the same time, I would like to emphasize that no article is from my country and I do not personally know any of these authors. This recommendation is supremely determined by the knowledge of renowned publications on the subject:
- Guthrie, W. S., Cooley, D., & Eggett, D. L. (2007). Effects of reclaimed asphalt pavement on mechanical properties of base materials. Transportation Research Record, 2005(1), 44-52.
- Tao, M., & Mallick, R. B. (2009). Effects of warm-mix asphalt additives on workability and mechanical properties of reclaimed asphalt pavement material. Transportation Research Record, 2126(1), 151-160. (USA)
- Copeland, A. (2011). Reclaimed asphalt pavement in asphalt mixtures: State of the practice(No. FHWA-HRT-11-021). United States. Federal Highway Administration. Office of Research, Development, and Technology.
- Hoy, M., Horpibulsuk, S., & Arulrajah, A. (2016). Strength development of Recycled Asphalt Pavement–Fly ash geopolymer as a road construction material. Construction and Building Materials, 117, 209-219. (Thailand)
- Farooq, M. A., & Mir, M. S. (2017). Use of reclaimed asphalt pavement (RAP) in warm mix asphalt (WMA) pavements: A review. Innovative Infrastructure Solutions, 2(1), 1-9.
- Guo, M., Liu, H., Jiao, Y., Mo, L., Tan, Y., Wang, D., & Liang, M. (2020). Effect of WMA-RAP technology on pavement performance of asphalt mixture: A state-of-the-art review. Journal of Cleaner Production, 266, 121704.
- Yousefi, A., Behnood, A., Nowruzi, A., & Haghshenas, H. (2021). Performance evaluation of asphalt mixtures containing warm mix asphalt (WMA) additives and reclaimed asphalt pavement (RAP). Construction and Building Materials, 268, 121200.
LNSA 149... Figure 2. Classification methods for recycling asphalt pavement processes... I recommend improving the image so that the texts in the image have approximately the same spaces from the edge of the text box.
LNSA 211...Table 2. Specifications and guidelines for using RAP [56-64]...I recommend adding abbreviations and designations of variables (including the physical units considered) to the table, which are not directly mentioned in the article.
LNSA 245-248...Hence, 35% of RAP in the asphalt mixtures could reduce fatigue as the strain level increases. Furthermore, thermal cracking could occur in cold temperatures due to continuous loading on the asphalt pavement [76]... I recommend identifying or quantifying "cold temperatures" in more detail. It would also be useful to indicate in the text what the average and maximum daily air temperatures are in Oman or the Persian Gulf.
LNSA 304-305... The research decided the adequacy of the utilization of frothed asphalt– balanced out recovered asphalt from full-profundity recovery (FAS-FDR) as the base material for adaptable asphalts... It would be useful to format the sentence so that the abbreviation (FAS-FDR) is not divided into 2 lines and explain the abbreviation used.
LNSA 349-360...7.2. A case study in European countries... It would be appropriate to add other European countries, as the technology in question is a frequent pavement rehabilitation technology in the EU countries.
In my opinion, the overview of research articles contributes to the scientific credibility of the mentioned environmental technology in the field of pavement engineering in the Gulf countries. The assessed article "Towards Encouragement Sustainable Utilization of the RAP Materials for Pavement Construction in Oman: A Review " can be assessed as good, containing certain shortcomings that need to be eliminated. From the aspect of quality of reviewed article, in case of incorporation of comments, or relevant justification of their non-incorporation, I am able to process a repeated review within 3 days.
Author Response

(The authors gave the same response as above.)

Reviewer 3 Report
The authors reviewed the usage of RAP in asphalt pavements in order to create a framework for practical usage of RAP to build sustainable pavement systems. The content was basically comprehensive. However, there are some problems existing in this paper which the authors must pay attention to deal with.
- The title of the section 2 is suggested to be modified.
- Please offer the full name for the term which is mentioned at the first time. For example, the WMA and HMA in section 2.
- Some expressions in the manuscript are not clear. An English native speaker is suggested to carefully proofread it again.
- Some gramma or format mistakes can be found as well. Please carefully check through the manuscript. For example, CO2 in the abstract.
- The aspect ratio of the words in Figure 1 seems strange.
- The caption of the Table 1 should be modified.
- Please pay more attention to the literature from Asian researchers, for example, when the authors summarized the past research on various RAP methods.
- The section 6 “Characteristics of Incorporating RAP” should be supplemented by the researches with respect to experimental tests and numerical simulation.
- Is there any case study in Oman about the application of the RAP?
- The conclusions and outlook are suggested to be clearly separated in the last section.
- The format of the reference is not consistent.
Author Response

(The authors gave the same response as above.)

Round 2
Reviewer 2 Report
By the author of the article "Encouraging Sustainable Use of RAP Materials for Pavement Construction in Oman: A Review" (original title Recycling Towards Encouragement Sustainable Utilization of the RAP Materials for Pavement Construction in Oman: A Review), I would like to present this summary evaluation. I am very pleased to be able to sincerely congratulate the authors on the excellent quality of the second version of their scientific contribution. The authors also took great care to incorporate my mandatory and facultative recommendations, although they certainly took a considerable amount of time. I sincerely thank the authors for the opportunity to expand my territorially limited knowledge and familiarize themselves with the new findings presented in the second version of the article under consideration. I keep my fingers crossed for publishers in sustainable improvement of their very useful magazine, even for my scientific field of holistic perception of sustainable pavement design, construction, management and recycling. In conclusion, I have no choice but to agree with the authors, that The Sultanate of Oman created the real conditions for implementation environmental-friendly for ls of the Oman 2040 vision in the pavement engineering too. The scientific approaches presented by the authors are fully convergent with my efforts, which the primary concern is preserving natural resources and reducing the impact of COâ‚‚ emissions into the environment achieved through holistic pavement engineering.
In conclusion, I would like to abstract the presented facts into repeated thank and congratulations to the authors as well as the publisher.
Author Response
Thank you for your positive, fruitful comments and suggestions, which have improved the quality of our manuscript.
Reviewer 3 Report
I am pleased that the questions were carefully answered by authors, and the suggestions were all considered by the authors according to the feedback.
Author Response

(The authors gave the same response as above.)
